# The Impact of Social Media on Risk Communication of Disasters—A Comparative Study Based on *Sina* Weibo Blogs Related to Tianjin Explosion and Typhoon Pigeon

**DOI:** 10.3390/ijerph17030883

**Published:** 2020-01-31

**Authors:** Tiezhong Liu, Huyuan Zhang, Hubo Zhang

**Affiliations:** 1School of Management and Economics, Beijing Institute of Technology, Beijing 100081, China; liutiezhong@bit.edu.cn (T.L.); 18813111238@163.com (H.Z.); 2China Electronics Standardization Institute, Beijing 100007, China

**Keywords:** risk communication, man-made disaster, natural disaster, social media, social network analysis

## Abstract

Social media has brought opportunities and challenges to risk communication of disasters by undermining the monopoly of traditional news media. This paper took blogs about Tianjin Explosion and Typhoon Pigeon posted through *Sina* Weibo as empirical objects. Moreover, the paper used the analytical method of social network to conduct a comparative study on the network structures of information disseminated among different types of disasters, with the goal of uncovering the impact of social media on different types of risk communication of disasters. The result shows a different impact of the risk communication on the two types of disasters. While the role of social media for the risk communication of natural disasters is mainly to influence information dissemination, the roles of social media for the risk communication of man-made disasters are to transmit information as well as to communicate emotions. The differences seen within the structure of social media networks are causes differences in functions. Specifically, the structure for the social media communication network on man-made disasters takes on a “core - periphery structure” which is endowed with both information communication and emotional communication functions. Also, the role of the opinion leaders for the subnet is found to be significant while the communication within small groups is kept pretty active; additionally, the slow speed of information transmission of the network could result in easily distorted information. On top of that, the network is characterized with intense vulnerability to the attacks on core nodes. In contrast, the social media network for natural disaster risk communication is not seen with an obvious “peripheral-core” structure which is a relatively pure information transmission network with relatively equal principal status. In other words, the entire network is found with stronger connectivity and relatively faster information transmission speed. Furthermore, the nodes inside the network are found to have weaker control over information transmission. In sum, the research results are helpful in improving the risk communication theory based on social relations, optimizing the communication structure of disaster information so as to change the effect of risk communication.

## 1. Introduction

Social media can significantly impact the information exchanges with its simple, fast and powerful cohesion force. According to statistics, as of June 2017, there were 751 million Internet users in China, with the overwhelming majority of information obtained by social media [1]. The certain advantages of the speed of information dissemination and appeal of social media [2] have brought opportunities for risk communication of disasters. For example, the Red Cross received $8 million donations within 48 h after the 2010 Haiti Earthquake, thanks to the relevant blogs and pictures posted through social media [3]. The social media has broken the monopoly of traditional news media on information dissemination by letting people create their own online blogs. Meanwhile, as online information is prone to be distorted, it has also brought risks to communication on disasters [4]. For example, some scholars have found, in their studies of rumors, that the reasons why rumors on common knowledge have been flowing consistently in social media is due to the large number of audiences and the difficulties to refute them [5]. In addition, the complexity of the causes of man-made disaster has made the official information untrustworthy to the public [6], which led social media become a greater contributor of disaster information. It is fair to say that, with the public’s increasing interest towards social media, an extended research on different characteristics of risk communication of disasters must be carried out to enable us to take better advantage of the positive impact of social media on risk communication of disasters, while minimizing the negative impact of social media in a timely manner.

Many relevant researches have been carried on this kind of topic. King J., Jones N. studied the impact of social media on risk communication from the perspective of interconnected scenario and information dissemination mechanism, and found that is the impact of social media is less important than traditional media [7]. What is more, FEMA has actively used various social media to provide disaster related information to the public [8]. Different strength of social relationships formed by different social media has produced different effect for risk communication. For example, It has been found that while the online relationship in Facebook is to some extent based on the offline relationship [9], most relationships formed in twitter, Weibo, etc. are one-way, asymmetric connections [10]. As another example, it has been demonstrated by Prof. Granovett that a weak tie denotes a weak relation between individuals, which offers more room for the transmission of new information and a variety of resources; a strong tie denotes a relationship between individuals with stronger relations, which could produce greater redundancy problem in information transmission as well as offer some psychological support, such as trust, respect and sense of responsibility [11]. This is obvious that the communication network formed through social media is a weak relationship network, which possesses such characteristics as large-scaled, diversified, etc. Moreover, more information can be acquired through social media [12]. It can be said therefore, that the social relations, as shown from social media network, play important roles in risk communication of disasters.

This study will use the method of social network analysis to study the social relations formed by social media. Based on the comparative analysis on natural disaster and man-made disaster, the impact of the network structure of social media on risk communication of disasters will be discussed. Moreover, the research results will help solve such practical problems, as well as some theoretical problems related to risk communication based on social relations.

## 2. Materials and Methods

### 2.1. Theoretical Framework

#### 2.1.1. Circulation Pattern of Information

The technical complexity of man-made disaster which causes an asymmetric distribution of disaster information has made it hard for the general public to acquire first-hand information, causing the public to find various means to obtain disaster information, means such as the official channels, social media, or oral accounts from friends, etc. [13]. In addition, the stigmatization, responsibility distribution and other characteristics of man-made disaster [14] can lead to: complex interest distributions; the formation of different interest groups and antagonistic emotions, which has put much emphasize on the role of opinion leaders [15]. Based on the different characteristics of natural disaster and man-made disaster, a disaster information flow pattern in social media has been found, as shown in Figure 1.

In Figure 1, S represents the source of information, which contains natural and social clues; C represents the original subject of the social media-Weibo; C_A_ is the subject of A class blogs, which includes blogs posted by either governmental or media; C_B_ symbolizes the subject of B Class blogs, which includes blogs of unofficial organizations and individuals; “O” represents the opinion leaders and “P” is short for “the public”.

First of all, for risk communication on man-made disaster is influenced greatly by opinion leaders, the information was processed and filtered several times. For example, the hazardous information on chemical facilities can only reach to the public through the path of “enterprise examiner—higher government examiner—grass-root government examiner”. To summarize, the following characteristics has been found for the man-made disaster: to start with, the source of information is diverse, which enables the information receivers-CA, CB, O obtain relevant information through a variety of natural clues and social clues. Next, as the opinion leaders are formed virtually, the information feedbacks among core groups CA, CB and O are relatively sufficient [16]. In addition, the guidance of opinion leaders to the subjects of the small groups is reflected by the fact that the opinion leaders can guide, to some extent, the flowing direction of the mainstream information within their small groups; most information that opinion leaders disseminate to the public flows in one-way. Finally, the characteristic differences among different small groups can lead to different opinions within different small groups.

Next, unlike man-made disaster, there is rarely any conflict among different interest groups within the natural disaster communication groups. For instance, individuals can communicate directly with each other in a relatively equally and smooth way, which tempers opinion conflict and then weakens the role of opinion leaders [17]. Hence, the biggest difference between the information flow of natural disaster and the information flow of man-made disaster lies in the role of opinion leaders, as shown in Figure 1. In cases of risk communication on natural disaster, the lack of antagonistic interests groups has led to a lack of conflicting views as well as corresponding opinion leaders inside small groups; on the other hand, the information obtained by the public on natural disaster is relatively objective, which results in relatively fewer opinion information, especially the opposing opinion information. 

#### 2.1.2. The Risk Communication Model

Based on the above analysis, a conceptual model of the impact of social network on risk communication is established as shown in Figure 2. The model shows different network characteristics between the social media of two kinds of risk communication of disasters, which can impact the effect of risk communication. In addition, for different network entities, the impact can be different depending on the different locations of the opinion leaders or the groups’ preferences. What is more, the topic preference of the public can also exert an impact on information dissemination [18,19].

### 2.2. Data Source

Social media covers such social network sites, as Weibo, Wechat, Online social forums and other types of forums. In this study, certain information in the authoritative *Sina* Weibo (hereinafter referred to as “*sina*”) forum has been selected as the data source. According to the statistics, as of March, 2018, more than 800 million people have registered *Sina* accounts, which include more than 175,000 governmental accounts. As another example, more than 100 million pieces of information would be posted through *Sina* every day, which covers all aspects of social news and hot topics [20]. In addition, the threshold for blogs posted through *Sina* is set lowly: users can, as they like, forward all kinds of interested news, disseminate knowledge or express their mood through either blogs, pictures, videos, links or other forms of communication. Moreover, the fact that the information can be posted or searched at any place, any time has enabled the information flow or exchanged at a faster rate [21]”. 

On the one hand, to study the impact of social media on risk communication on man-made disaster, the explosion in Tianjin Binhai New Area” (hereinafter referred to as “Tianjin Explosion”) has been selected. The accident was an extremely serious production safety liability accident happening on 12 August 2015 which has brought great social impact. Since the Explosion, there was a lot of information posted on *Sina* on the accident, making the accident a hot topic of Weibo. Therefore, this study will use the discussion on Tianjin Explosion on *Sina* as the research object, focusing on: the characteristics of social network formed by risk communication on hazardous chemicals in social media; the impact of characteristics of social network on risk communication on hazardous chemicals. According to the index data of search engine of Baidu, related information posted through Weibo was mainly concentrated in the first half month of the accident. Therefore, the relevant content on *Sina* have been retrieved from 12 August 2015 to 31 August 2015.

More specifically, an advanced search function provided by *Sina* was used to collect data, which has set such key words as: “Tianjin”, “Tanggu”, “Tianjin Port”, “Binhai New Area”, “explosion”, “fire”, “accident”, “8.12”, etc. (data collected on 17 July 2018). After preliminary screening of data, data unrelated to the Tianjin explosion was manually excluded. Further, as many blogs on the accident were not much forwarded, commented or “liked”, yielding no big social impact, the research objects would only include those that have at least one date involving more than 1000 individuals. Finally, 562 blogs posted through 396 Weibo accounts have been singled out. Among them, 17 were posted through government; 185 were posted through news media of Weibo; 283 were posted through either celebrities or non-celebrities; 77 were posted through unofficial organization in Weibo (such as Star support groups and interest groups). In total, 417 blogs were posted through certified accounts of Weibo, and 145 were posted through non-certified accounts.

On the other hand, for the study on the impact of social media on risk communication on natural disaster, Typhoon Pigeon has been selected as a typical study object. Typhoon Pigeon was a natural disaster formed on 20 August 2017. The disaster has brought huge destruction in areas of Zhuhai, Hong Kong, Macao and other other southern China, causing 24 deaths and 6.82 billion worth of US dollars on direct economic losses. According to index of Baidu search engine, relevant information concerning this typhoon was published mainly in the first ten days of typhoon formation. Therefore, the relevant blogs in Weibo of this typhoon from 20 August 2017 to 31 August 2017 have been retrieved.

Using similar search method on Weibo blogs on Tianjin Explosion, the key words “typhoon pigeon”, “pigeon”, “Zhuhai typhoon” as keywords has been chosen. Finally, 70 Weibo blogs posted through 51 Weibo accounts were screened out: Among them, 6 were posts of the government; 56 were from news media of Weibo; 3 were created by celebrities or non-celebrities; 5 were posted through unofficial organizations in Weibo (such as various Star support groups); 65 were issued by certified account, and 5 were issued by non-certified account.

### 2.3. Research Methods

The social network analysis method was selected to analyze and create models. The forwarding network of Weibo has been analyzed to explore the characteristics of social media network. The following steps have been used to build the online risk communication on Tianjin Explosion and Typhoon Pigeon [22]:

First of all, taking *Sina* Hot Blogs as an actor, a N group which covers all g actors was set up, which is recorded as $N=\left\{n_{1},n_{2},\cdots n_{i},\cdots n_{g}\right\}$. If there is a relationship between the pair of actors, and the relationship is directional, the pair can then be thought as an orderly pair $\langle n_{i},n_{j}\rangle$, and can be recorded as: $n_{i}\to n_{j}$. If a forwarding relationship was found between the two *Sina* accounts, the two will be considered as orderly pairs, with a forwarding direction from one account to the other. To record all forwarding relationships of accounts of *Sina* Hot Blogs as orderly pairs, the Forwarding Adjacency Matrices of Hot *Sina* Blogs were constructed. Moreover, because the Matrices are directional, they are formed asymmetrically. The Forwarding Adjacency Matrices for Tianjin Explosion and Typhoon Pigeon were constructed respectively (omitted here).

Then, the Forwarding Adjacency Matrices among *Sina* Hot Blogs were converted into pajek files. Further, the method of Clustering Placement was used for visual display, that is, referring to the nature of the nodes and the connections among them, the nodes of the networks were clustered by Kamada Kawai algorithm. Hence, the forwarding network for Hot Blogs on Tianjin Explosion (hereinafter referred to as “Tianjin Explosion Network”, as shown in Figure 3a) and the forwarding network for Hot Blogs on Typhoon Pigeon (hereinafter referred to as “the General Explosion Network for Typhoon Pigeon”, as shown in Figure 3b) were constructed.

The following can be generalized from Figure 3: (1) the Tianjin Explosion Network has a relatively obvious “core-periphery” structure, which shows a somewhat star-shaped structure with some of the network’s nodes even drifted away from the other parts of the network; (2) the General Explosion Network for Typhoon Pigeon is relatively evenly structured with good connectivity; all nodes have the same information transmission capacity.

## 3. Results

### 3.1. Impact of the Overall Network on Information Dissemination

#### 3.1.1. Impact of Characteristics of the Small World on Information Dissemination

The two basic concepts depicting characteristics of the small world are clustering coefficient and characteristic path length. To conclude, all small worlds must have large average clustering coefficient and very small average path distance [23,24].

Parameter 1: Clustering Coefficient. The clustering coefficient is the average value of network density, which can reflect, to some extent, the level of overlapping between individual networks in social network. Further, the concept can be held as a characteristic index for network structure. The clustering coefficients of Tianjin Explosion Network and Typhoon Pigeon Network were 0.280 and 0.599, meaning that the proportions of actual correlations of numbers of edges to the maximum possible correlations of numbers of edges between any adjacent nodes for the two networks were 28.0% and 59.9% respectively, indicating a relatively smooth information communication locally. Having said that, the clustering coefficient of Tianjin Explosion Network was smaller than that of Typhoon Pigeon network, which suggests that the relationship of the nodes of Tianjin Explosion Network was kept relatively loosely; the nodes of Typhoon Pigeon Network were bonded closely at last, the possibility of effective communication between different nodes in Typhoon Pigeon network was big.

Parameter 2: Average Path Length. The average path length can directly reflect the transmission efficiency of social media network. The shorter the average path length is, the faster the information can transmit between different nodes in the network. The nodes of Tianjin Explosion Network and Typhoon Pigeon Network were 396 and 51 respectively, with corresponding average path length of 2.882 and 1.829 respectively. It shows that the average path length of Typhoon Pigeon Network was shorter than that of Tianjin Explosion Network, indicating a faster transmission speed for Typhoon Pigeon and a slower transmission speed for Tianjin Explosion. The speed of information transmission has been slowed by the long information transmission chain, which points to greater possibility on rumor generation.

Both networks hadshorter average path lengths and higher clustering coefficients, displaying evident characteristics of “small world network”, which can help speed up information dissemination.

#### 3.1.2. Impact of Basic Network Characteristics on Information Dissemination

From the point of view of social network analysis, the indicators of network degree distribution, network diameter, network density, network correlation degree [25,26,27,28] are normally selected for the analysis on network feature.

Parameter 1: Average out-degree. The average out-degree can reflect the breadth of information sources, that is, the larger the average out-degree, the more likely the information can be obtained by the public. Also, the psychological activities of the public on disaster information can also be inferred. While the average out-degree of Tianjin Explosion Network was 23.96, the average out-degree of Typhoon Pigeon Network was 32.5, indicating a more possibility for information transmission on Typhoon Pigeon. It can be said then that: Although the number of nodes in Tianjin Explosion Network is significantly more than that of Typhoon Pigeon Network, its average out-degree is less than that of Typhoon Pigeon Network, which on some levels, can reflect the public’s distrust on information on man-made disaster and their intention to pass down information flown solely from trusted sources, or from those opinion leaders who hold discourse power in corresponding fields.

Parameter 2: Network Diameter. The network diameter refers to the maximum value of the shortest path between any two nodes in the network, which is generally used to measure the network connectivity. It can be used to calculate the coverage of the information and its distortion level. For this paper in particular, the network diameter of Tianjin Explosion Network and Typhoon Pigeon Network were calculated as 9 and 5 respectively, indicating a long information transmission chain. The long chain has led to a slow information transmission speed as well as a possibility of rumor dissemination [29].

The social media was found to have a slow propagation rate and a high distortion rate on information, which could have been caused by the division of “factions” inside the Tianjin Explosion Network. That is, because of the sensitivity and conflicts of interests involved, subjects of Weibo were more willing to forward information passed down by their “own people”. It has caused difficulty for the communication between different factions, making communication paths between individuals of different factions longer, which would be easily deduced through further analysis on the forwarding relationship of “Sina Family”.

Parameter 3: Network Density. Network density is the ratio between the actual number of edges and the theoretical maximum number of edges. In particular, the network density describes the average level of relations of nodes of the social networks. That is, the closer the nodes are connected, the greater the network density is. In the case of Tianjin Explosion Network, the density of the Network was calculated as 0.0303, indicating a low level of association and sufficiency of information among nodes. In the case of Typhoon Pigeon network, the network density was calculated as 0.3183, indicating a high level of relation and information exchanges among nodes.

In short, the Tianjin Explosive Network can be held as a low density network and which was symbolized with bigger clustering coefficient. Moreover, it is difficult to conduct long-distance information communication within this type of social media network. Besides, the communication in small groups can be said as relatively active.

Parameter 4: Network Relevancy. Network relevancy refers to the proportion of relevancy between any two nodes in a directional network. The higher the network relevancy is, the greater the possibility for information to travel between these nodes in the network is. In view of this paper, the network relevancy of Tianjin Explosion Network and Typhoon Pigeon Network were calculated as 0.1633 and 0.3528 respectively.

The relatively low network relevancy of Tianjin Explosion Network indicates: there were many lonely information islands in the information dissemination and more one-way communication; the collection of communication feedbacks has become harder as information cannot be received by groups on time; the relatively high network relevancy of Typhoon Pigeon reflects that there were just a few lonely information islands in its information dissemination process; the proportion of two-way communication on the network was relatively high; the equal status of individual subjects ensured the coverage rate of information and efficiency of feedback collection.

#### 3.1.3. The Influence of Subgroup’ Characteristics on Information Dissemination

As some nodes in the network were more closely related, they form subgroup with certain characteristics, which could be called as “cohesive subgroup” (referred to as “subgroup”). An analysis on network of subgroups can show clearly: the internal structures of the networks; the common characteristics of the nodes under the same subgroup and characteristic of relationships among different subgroups [30].

(1) Analysis on the Characteristic of the Subgroup of Tianjin Explosion Network

Using Louvain algorithm, Tianjin Explosion Network have been divided into 26 subgroups. In order to form intuitive understandings of the subgroups, initially, the nodes belonging to the same subgroup were classified by the method of Shrinkage Subnets [10]. The specific operation steps can be described as below: for the first step, all nodes belonging to the same subgroup were shrunk to one particular subgroup node. What is more, in order to ensure that connections between different subgroups are not too loose, it can be stipulated that a connection can be established only when the original connections between the subgroups are greater than 20. At last, a subgroup’ network has been obtained, as shown in Figure 4.

Each node in the graph represents a subgroup, and the number of original nodes in the subgroup varied from 1 to 155. Among them, subgroup A contained 155 nodes, subgroup B contained 64 nodes, subgroup C contained 49 nodes, subgroup D contained 45 nodes, subgroup E contained 36 nodes, and subgroup F contained 26 nodes. Moreover, all other subgroups not have shown here contained only one node. The characteristics of internal nodes of subgroups A, B, C, D, and E were discussed below to form better understandings on compositions of different subgroups as well as the formation causes of relationships among subgroups. Furthermore, the internal structures of subgroups A, B, C, D, E and F were displayed and analyzed, respectively, through the subnet extraction function of Pajek software. Finally, the structures of these subgroups were shown correspondingly as Picture A, B, C, D, E, and F in Figure 5.

On the characteristics of subgroup A: this is an important subgroup, which was related to subgroup B, C, D and E. Within the groups, subgroup A transmitted information to subgroup B, D and E as well as obtained information from subgroup B, C and D. However, the objects from which subgroup A receives its information or pass its information were not exactly the same, which indicates that the directions were not two-way for some information transmitted among different subgroups.

On the internal structure of subgroup A: the subgroup was mainly comprised with Weibo blogs published by celebrities and other unofficial organizations. In addition, it was found that 686 relationships were formed among 155 nodes of subgroup A; the average out-degree was calculated as 8.74 and the network density was calculated as 0.0278. Also, the nodes within this group were loosely connected; there was quite obvious difference among the core and peripheral nodes, that is, while some nodes were more closely connected to other nodes, there were other nodes that have fewer connected nodes.

On the characteristics of subgroup B: although the number of internal nodes of subgroup B was a lot less than that of subgroup A, similar to subgroup A, C, D, E and F, subgroup B was formed by two-way connections. They all acted as “bridges” between subgroup F and the other four subgroups. Obviously, subgroup B was at the center of the network and so is very important for information transmission.

The internal structure of subgroup B shows: the subjects of Weibo accounts in subgroup B mainly included governmental departments dealing the explosion accident in Tianjin and the news media which releases timely information to the public. There were 1280 connections found among the 64 nodes of subgroup B, with an average out-degree of 36.06 and a network density of 0.2575. The node was closely related to one another, indicating that a large number of Weibo blogs has been forwarded with high level of risk communication. Compared with subgroup A, there was no significant difference between the core and peripheral nodes of subgroup B.

Third, on the characteristics of subgroup C and D: similar to subgroup A, the subjects sending or obtaining the information were inconsistent.

The internal structure of subgroup C can be concluded as: the subjects of two Weibo accounts at the core position of subgroup C were CCTV News and People’s Daily, the two state-level news media, while the Weibo accounts on the periphery were owned basically by non-celebrities or the general public or individuals. Among them, 60 relationships were established among 49 nodes of subgroup C, with the average out-degree of 3.33 and the network density of 0.0476. The relationships between the nodes were quite loose. Not only that, the subgroup C was formed by a very obvious “core-periphery” structure, a star-shaped network, which served as a bonus to the control of information accuracy.

The internal structure of subgroup D can be described as: the core layer of Subgroup D can be characterized as “*Sina* Family”, which included such “members” as *Sina* Tianjin, *Sina* Shandong, *Sina* Henan, etc. This “*Sina* Family” presented as an important information source for Tianjin Explosion. Also, the relationship of information transmission among subjects of the system was relatively equal and smooth. The peripheral layer was formed by the non-celebrities or individuals of Weibo accounts. 231 relations were found between the 45 nodes of subgroup D, with an average out-degree of 12.83 and a network density of 0.1833. The relationship among the nodes was relatively close. Moreover, there are evident hierarchical characteristics inside subgroup D. It can be concluded then: the core layers were formed by inner layers; there were not too much position differentiation for the nodes in the core layers; the outer layers were the peripheral layers.

On the characteristics of subgroup E: the characteristic of subgroup E was found to be quite different from other subgroups. While it obtained information from subgroup A and B, it did not transmit information to other subgroups. This indicates that while individuals inside subgroup E were willing to forward Weibo blogs to other subgroups, the individuals of other subgroups did not attach great importance to the information forwarded by individuals of subgroup E.

The internal structure of subgroup E can be concluded as: the subjects of Weibo accounts of subgroup E were often called as the “little celebrities” who were famous within their own respective communities or particular fields. A total of 190 relationships were established between the 36 nodes of subgroup E, which had an average out-degree of 7.60 and a network density of 0.0776. The nodes were relatively loosely connected. Relatively speaking, the boundary between the core layers and the peripheral layers of the subgroup can be quite fuzzy, with little geographical differences among subjects of the Weibo accounts.

Fifth, the characteristics of isolated subgroups are: many isolated subgroups in the network were owned by non-celebrities, i.e., individual users of Weibo accounts. It can be seen from this graph that the number of “likes” obtained by these nodes was far more than the number of blogs that were forwarded. This reflects that although Weibo readers echo with information transmitted by non-celebrities, considering the limited influence of the authors of the blogs, in reality, the motivation to transmit the information of the blogs is just not there. 

The main characteristics of internal structure of isolated subgroups can be generalized as: most isolated subgroups were owned by non-celebrities, that is individual subjects of Weibo accounts. The characteristic of the isolated subgroup F can be generalized as follows: 91 relationships were found among the 26 nodes; the average out-degree of this subgroup was 7.00 with network density of 0.1346; the relationships between the nodes of this subgroup was relatively close; there was no significant geographic difference between the core layers and the peripheral layers of subgroup F.

The subgroup of Tianjin Explosive Network were classified mainly based on the nature of the subjects of the accounts, which is to say that: The identities of the subjects and the subjects’ common interest were key factors that can decide which groups the subjects of Weibo accounts belong to; risk communication was smother within the same group; the structure of subgroup of Tianjin Explosive Network was quite distinctive, meaning the characteristics of risk communication and its transmission forms for man-made disaster were diverse and complex; while the official subjects appeared more at the core of the network, the public was seen more obviously on the periphery of the network.

(2) Analysis on the Characteristics for the Cohesive Subgroup Network of Typhoon Pigeon

The structure of subgroup Typhoon Pigeon Network was analyzed and then divided into five subgroups. To analyze the characteristics of the cohesive network structure of Typhoon Pigeon and its possible influence on the risk information transmission, similar to the analysis process on Tianjin Explosion, the diagram of the subgroup structure of Typhoon Pigeon was obtained by extracting the network of the subgroups, as shown in Figure 6.

It can be generalized based on the network structures of Typhoon Pigeon’s subgroup: the type of structure of each subgroup network was relatively simple, with no obvious “core-periphery” structure; the structure was similar to the globally coupled network, that is, any two subjects in the network are directly connected with each other, with each subject having equal status in the network. These characteristic of this type of networks ensured a relatively fast and accurate information transmission and just a few information islands. Because the subjects can get information directly from one another, the network would be filled with confusing information and opposing views. On the other, the nature of the key nodes of the network display that the subject of the key nodes of Typhoon Pigeon subgroup’ networks was either media or the government, who, with their strong information identification ability, can ensure the consistency of information, and so can dissolve the issue of conflicting opinions to some extent. 

It can be said from a functional point of view that: (1) Typhoon Pigeon Network is a relatively pure information dissemination network; (2) The Tianjin Explosion network is not only an important network for information dissemination, but also an important network for emotional communication.

### 3.2. Impact of Centrality on Information Dissemination

#### 3.2.1. The Degree Distribution Analysis

The Degree Distribution refers to the degree distribution of the nodes in the network. A large number of research results show that the degree distribution of the actual network generally follows Power law distribution. Combined with the characteristics of social media network, the node’s out-degree has been selected for analysis. The out-degree of a node refers to the number of times the information of the node is forwarded by other nodes. The larger the out-degree is, the more the information of this node is forwarded by other nodes, and the more influential the information disseminated by this node might be. The degree distribution of Tianjin Explosion Network (out-degree) and the degree distribution of Typhoon Pigeon (out-degree) were drawn and shown in Figure 7.

On the one hand, most blogs on *Sina* Hot Topic of Tianjin Explosion, has been forwarded less than 20 times, with only a small number of blogs forwarded for more than 100 times, showing that the flow of information has followed the distinctive pattern of Power-Laws. Therefore, Tianjin Explosion Network had the “robustness” against “random failure” and “vulnerability” against “deliberate attacks” [31,32]. That is to say, even if a piece of false information appears randomly in the network, it is unlikely that it will affect the information accuracy of the whole network, but if the key nodes of the network deliberately or not deliberately publish false information, the information sent to the whole network will become untrustworthy. Therefore, the attention must be paid to the role of the key nodes in Weibo forwarding network so as to ensure the accuracy of key node information.

On the other hand, the Power Laws distribution was shown weakly on the out-degree distribution of Typhoon Pigeon Network, with relatively equal frequency among intervals, which proofs further on the lack of key nodes for the Typhoon Pigeon Network. The information stability of the network was strong, which means that, the probability of information distortion through one particular node was relatively small. Having said that, if you want to send out fast messages to the whole network through a particular node or clarify rumors through the node, the effect will not turn out to be good due to the lack of key nodes.

#### 3.2.2. Centrality and the Structural Holes

Centrality is a key index figure which can reflect the importance of a network node, which can be further characterized as Betweenness Centrality, Closeness Centrality and Weight Centrality [33,34]: the greater the value of Betweenness Centrality, the higher the possibility of the node acting as an intermediary as well as taking a greater control on the network resources; the greater the value of Closeness Centrality, the closer the node is to the center of the network; the Weight Centrality is used to weigh importance of one node from the perspective of the connection of this node to a key node. In addition, the Constraint Index of structural holes was introduced to examine the ability of a node in information and resources coordination. The smaller the value of the Constraint is, the more important the node is. The Centrality and values of the Constraint of Tianjin Explosion Network and Typhoon Pigeon Network were listed in Table 1.

It can be generalized that the centrality of the Typhoon Pigeon Network was generally greater than that of the Tianjin Explosion, which indicates that the nodes of the Typhoon Pigeon Network were more tightly connected, and more nodes were acting as the intermediaries and bridges during the process of information transmission. However, the constraints of the structural holes of Typhoon Pigeon Network were found to be smaller than that of Tianjin Explosion Network, which suggests that: the status of the nodes of Typhoon Pigeon network was relatively equal; the nodes had relatively weaker control on information transmission.

It appears at first that the structural differences of the social media on the two events have caused different characteristics of the networks. While the Tianjin Explosion Network was close to a scale-free network which contained certain key nodes, Typhoon Pigeon Network was more like a regular type of network; the constraints for the nodes of Tianjin Explosion Network were valued higher. Meanwhile, because of the small number of nodes in Typhoon Pigeon network, the possibility of each node acting as the intermediary or the bridge has been increased, which has produced generally bigger centrality figures. Through further analysis, it was found that during the accident of Tianjin Explosions, even when the public was enthusiastic about participating in the dissemination of relevant information and would love to carry out heated discussions, due to the social sensitivity of the causes of man-made disasters, the asymmetry of information and the authorities of information disseminators, the public was seen only on the periphery of information dissemination, resulting in a high constraint value for the overall network.

## 4. Discussion

It can be seen from the analysis towards two kinds of impacts of social media on different disasters.

For the social media network of man-made disaster, it shows core-periphery structure, as well as the greater control capability of key node. Considering the technical complexity, the social media network of this kind is endowed with both the function of information dissemination and emotional communication. As a result, this type of social media network presents the characteristics of factional divisions just like the actual social communities. Then, the opinion leaders of the networks and other key nodes within the networks would exert greater influence on the communication networks. Furthermore, while it is harder to conduct long-distance information communication the communication behavior inside the small groups can be relatively active. At the same time, for risk communication on this type of disasters relies heavily on key nodes, namely the opinion leaders, it would result in long information dissemination chains, which is prone to rumor generation. Finally, the speed of information transmission, as subjected to the abilities of the key nodes, is pretty slow. What’s more, such social media networks are often vulnerable to deliberate attacks. For example, if some opinion leaders make mistakes or deliberately spread rumors, it will more likely to cause panics to the network as a whole.

In view of the social media network of natural disaster, it can be concluded that: the social media network does not have a clear “periphery-core” structure and is a relatively pure information transmission network; the network presents regular characteristics and a stronger connectivity. For there are not many isolated information islands in the process of information dissemination, the proportion of two-way communication has become higher, which can ensure coverage and normal feedbacks of information. However, due to the lack of key nodes, the nodes are found to have relatively weak controls over information dissemination. That is, if someone wants to quickly disseminate certain information to the whole network, or to disseminate information through one particular note, the effect will not turn out to be so satisfactory.

## 5. Conclusions

To let the social media have better effect on risk communication as well as to avoid risk, the Tianjin Explosion and Typhoon Pigeon have been used as study objects. In particular, the discussion on the functional differences of social media on risk communication of man-made disasters and natural disasters has been carried out. It is shown at last that social media exerts different effect on each type of risk communications. While social media is mainly endowed with the function of information dissemination for risk communication of natural disasters, it plays both communication role and the role of emotional exchanges for risk communication network of man-made disasters. This can be concluded then that: differences of network structure can result in different functions of social media. In terms of man-made disasters, the cluster effect of social media is more obvious; moreover, the opinion is unified for individuals of the same subgroup; the information transmission is smooth; less information is transmitted between different subgroups; the antagonistic emotion exists inside each subgroup, which has caused the role of opinion leaders extremely important at this point. In contrast, for the natural disaster risk communication, differences are small for social media network structure; the network connectivity is stronger; information dissemination has become more efficient. Therefore, only the accuracy of the information would be considered for risk communication on natural disasters. What is more, the use of such networks should change from the primary type of “information dissemination” to the more advanced type of “knowledge dissemination”, disseminating not only basic information about the disasters but also deeper knowledge on how people should act in the face of such disasters. In respect of risk communication of man-made disasters, the aspects of information dissemination and emotional stability should be taken into consideration.Moreover, more focus should be paid on the construction of social relations rather than just information dissemination. The results can help improve the theory of risk communication based on social relations, optimizing the structure of disaster information transmission and changing the effect of risk communication.

## Figures and Tables

**Figure 1 ijerph-17-00883-f001:**
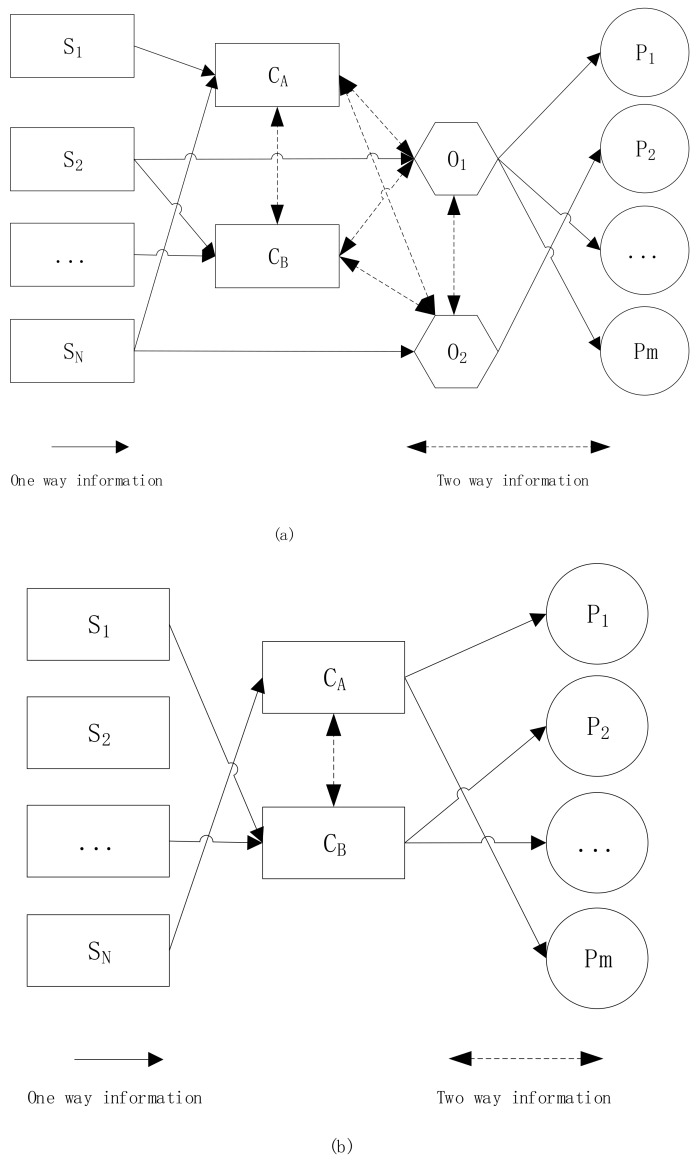
Flow Pattern of the Disaster Information in Social Media: (**a**) Information Flow Pattern of Man-made disaster; (**b**) Information Flow Pattern of Natural disaster.

**Figure 2 ijerph-17-00883-f002:**
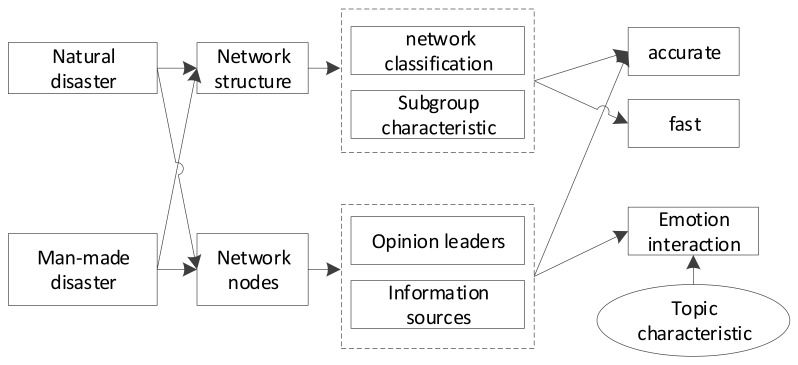
Conceptual Model on How Social Media Impacts Risk Communication.

**Figure 3 ijerph-17-00883-f003:**
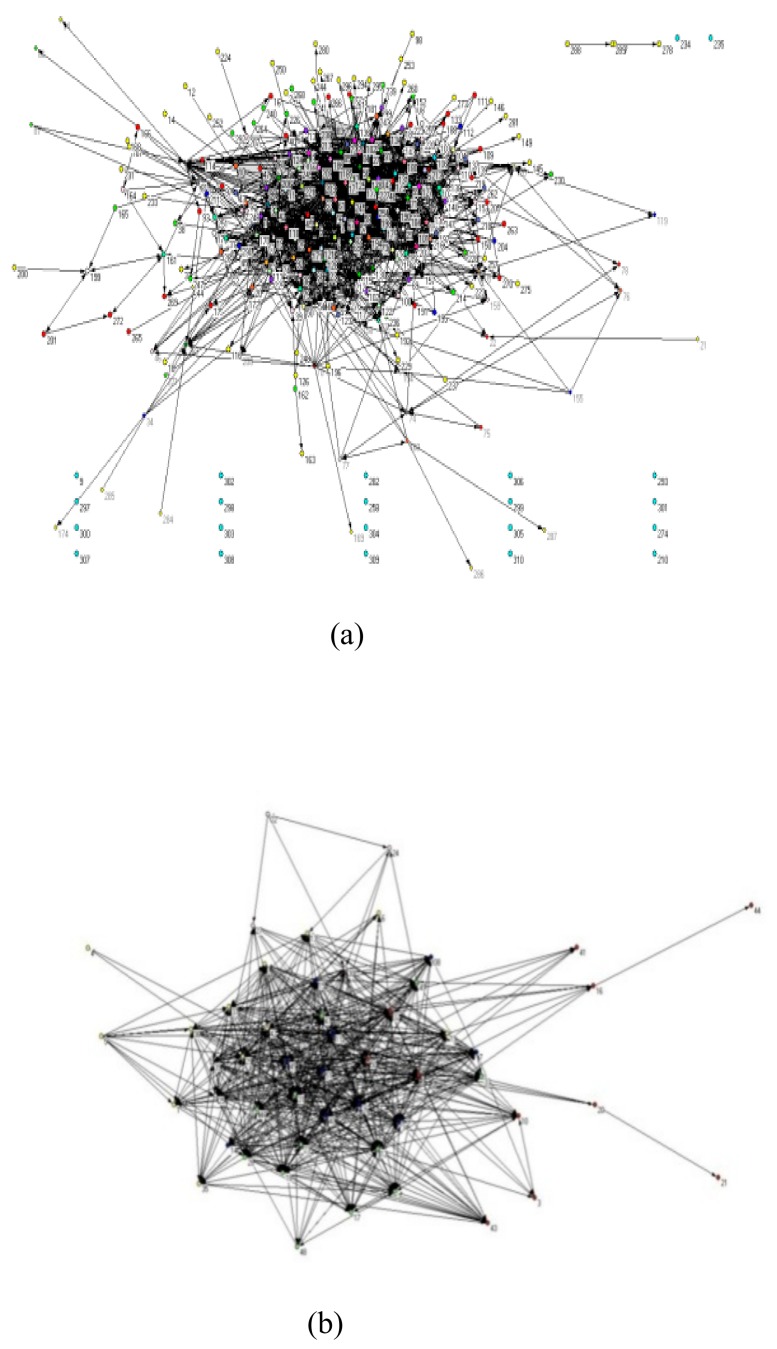
Features of Network Structure of Social Media: (**a**) the Forwarding Network for Hot Blogs on Tianjin Explosion; (**b**) the Forwarding Network for Hot Blogs on Typhoon Pigeon.

**Figure 4 ijerph-17-00883-f004:**
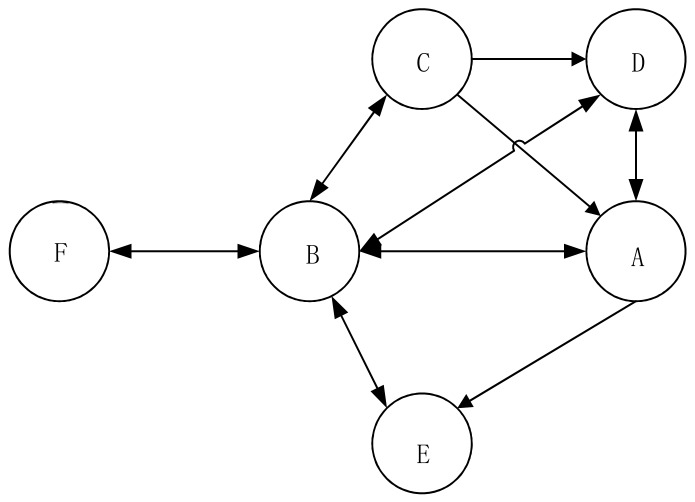
Shrinkage Network of Cohesive Subgroup of Tianjin Explosion.

**Figure 5 ijerph-17-00883-f005:**
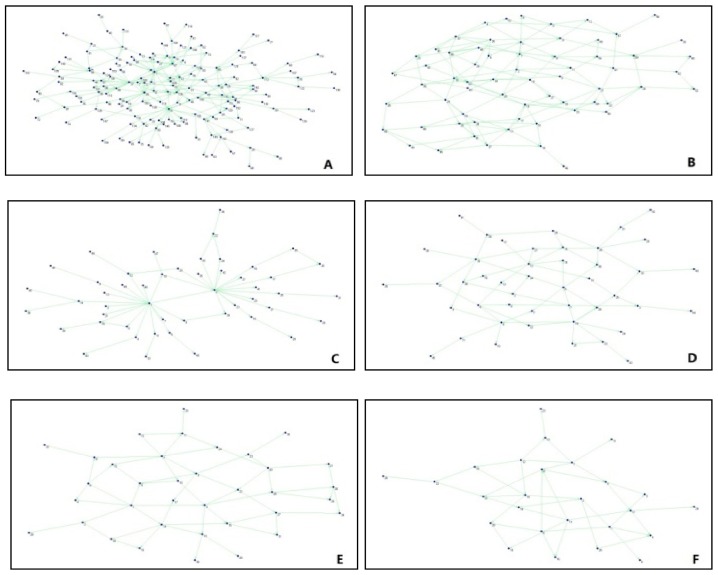
Network Structures of Cohesive Subgroup of Tianjin Explosion.

**Figure 6 ijerph-17-00883-f006:**
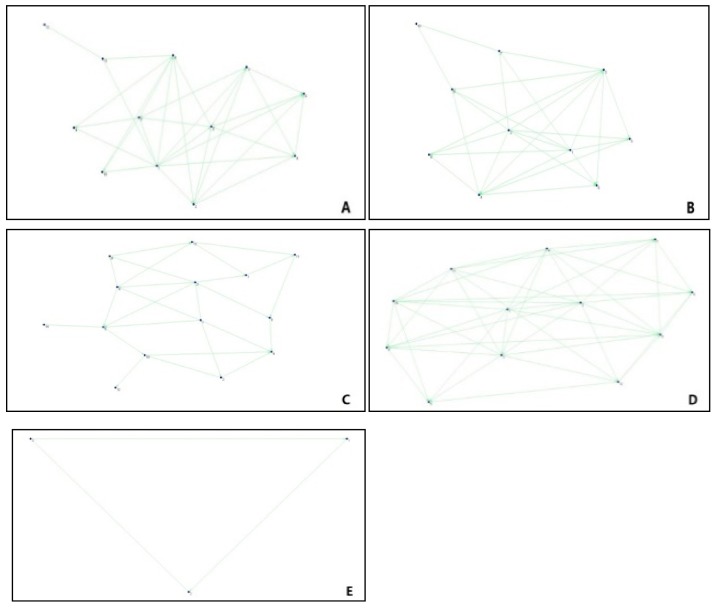
Network Structures of Cohesive Subgroup of Typhoon Pigeon.

**Figure 7 ijerph-17-00883-f007:**
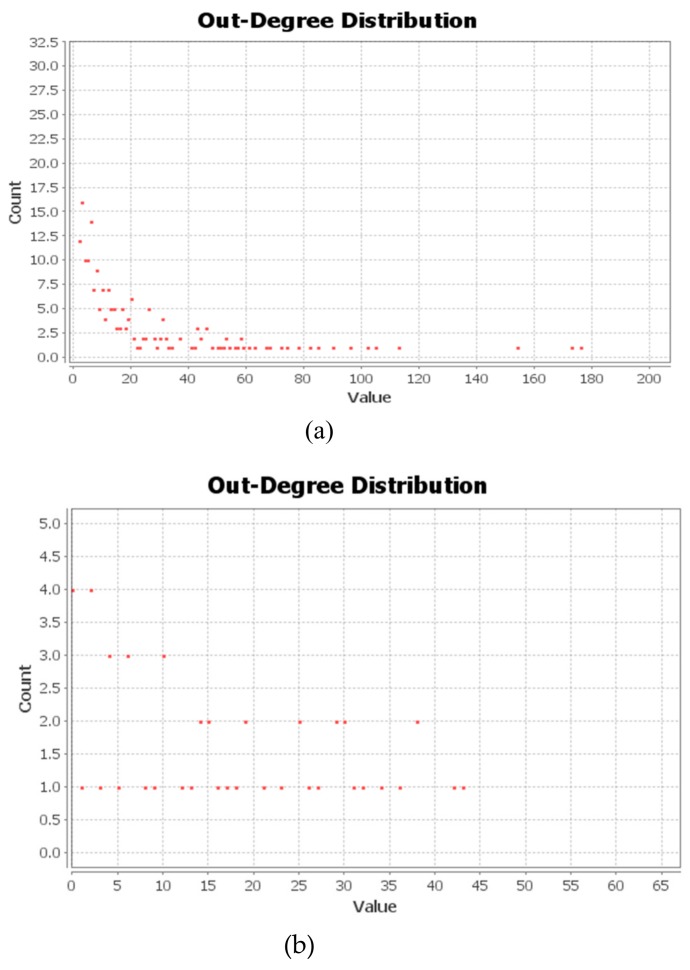
Degree Distribution of Social Media Network (out-degree): (**a**) Tianjin Explosion Network; (**b**) Typhoon Pigeon Network.

**Table 1 ijerph-17-00883-t001:** Comparison on the Centrality and Values of the Constraint.

	Tianjin Explosion Network				Typhoon Pigeon Network			
	Betweeness Centrality	Closeness Centrality	Weight Centrality	Constraint	Betweeness Centrality	Closeness Centrality	Weight Centrality	Constraint
Average	0.0024	0.1809	0.0319	0.2774	0.0153	0.5213	0.1241	0.1876
Standard	0.0053	0.1689	0.0389	0.3140	0.0183	0.1954	0.0648	0.1870
Mam.	0.0464	0.5552	0.2202	1.0000	0.0702	0.8734	0.2168	1.0000
Min.	0.0000	0.0000	0.0000	0.0230	0.0000	0.0000	0.0000	0.0994

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
