# Peer review of "The Impact of Social Media on Risk Communication of Disasters—A Comparative Study Based on Sina Weibo Blogs Related to Tianjin Explosion and Typhoon Pigeon"

_ijerph, 2020, doi:10.3390/ijerph17030883_

Round 1
Reviewer 1 Report
Overall, the article is well structured: it presents a solid introduction, discusses the main theoretical concepts and applies adequate research methods and techniques. The presentation of the results are also consistent.
The conclusion could be more extended and solid, explaining in short at was the objective of the study, its pertinence, the research methods and then the conclusions. The introduction and the conclusion of the article should be enough to understand the research.
Author Response
The section of the conclusion has since been re-organized and edited with careful considerations based on your suggestion. Mainly, the enhancement has been made on:
First, the overall structure of the Conclusion has been enhanced. In a word, the research purpose, data sources and research methods, and a summation about the main research conclusions have been inlcuded to improve the understanding of the readers on the paper.
Second, the practical significance of the research has been unravelled in the conclusion, which means that, for cases of communication on the risks of natural disasters in social media, only the accuracy of information needs to be taken into account when desseminating knowlege. However, when communicating risks of man-made disasters, the aspects of information dissemination and emotional stability have to be taken into consideration as well.

Reviewer 2 Report
This paper sets out to examine the role of blogs in communication about two disasters by examining how they influenced communication about those disasters through social networks.
It uses social network analysis to examine and compare the networks created in an anthropomorphic and a natural disaster.
The authors say their aim is to help improve social media use during disasters while also exploring theoretical ideas around risk communication and social relations.
The method, data analysis and presentation are all sound.
The discussion raises some important points for those organisations trying to communication in anthropomorphic and natural disasters and the conclusion is sound as well, making some strong points for those communicating about both types of disasters.
Overall the paper would benefit from a thorough edit as there are many awkwardly worded phrases.
In the introduction there is a mention of the Red Cross receiving donations after the Haiti earthquake – a reference is needed to support that.
Where there are references to traditional media – I think the authors are referring to traditional news media and that should be changed throughout.
Author Response
Comments and Suggestions1:Overall the paper would benefit from a thorough edit as there are many awkwardly worded phrases.
Response: We completely agree with your comment and have since made phrasal changes accordingly:
(1)The stiff sequential conjunction such as: first, second, third in the abstract and texts have been converted into more vivid conjunctions.
(2)Adjustment has been made on some punctuation and sentence patterns before and after the conjunctions, to make the meaning of the article clearer.
Comments and Suggestions 2:In the introduction there is a mention of the Red Cross receiving donations after the Haiti earthquake – a reference is needed to support that.
Response: To support relevant idea, the reference 3 on Haiti Earthquake has been added.
Comments and Suggestions 3:Where there are references to traditional media – I think the authors are referring to traditional news media and that should be changed throughout.
Response: We are grateful of your comment. After careful examination on the actual situation in China, the term “traditional media” which are shown in the paper has since been corrected as “traditional news media”.

Reviewer 3 Report
Dear Author,
Congratulations on a fine piece on research, I found it interesting and have made some questions and suggestions related to grammar, results, structure and content.
Many thanks for the opportunity to review.

Author Response
Comments and Suggestions:Congratulations on a fine piece on research, I found it interesting and have made some questions and suggestions related to grammar, results, structure and content.Responses:
(1)Line 22:Changes have been made on the Sequential Conjunctions;
(2)Line 33:Some modifiers have been replaced;
(3)Line 49: Some modifiers have been replaced;
(4)Line 52:
Response: More information has been offered for the references
(5)Line 86 and line87:“I find it strange that information flow diagram does not include flow directly between CA and CB. Various twitter feeds demonstrate retweeting of communication by the other? Could the public (P) not also be S?”
Response: The information flow in the graph has been moderated. In particular: 1) the information flow between CA and CB has been added; 2) while the Public (P) can be taken as information source (S), which contains C, O, P and other bodies, the P in the figure can be taken as the information receiver, or “the Public” in the risk communication process. Because of this, the two are different.
(6)Line 105: “Does it weaken the role or does it temper opinion or work as a measure of quality assurance”
Response: We completely agree with your comment and have since revised the expression as “which tempers opinion conflict and then weakens the role of opinion leaders”.
(7)Line 122:“Can you validate that social media impact on risk communication differs between man-made natural disasters? Intuitively one would think that there is commonality in communication between the two disaster groups.”
Response:That's a very good question. The emergence of social media has brought new and common changes to the risk communication of both natural and man-made disasters, such as a faster information transmission speed compared with the offline communication. However, an innate distinction has been found between risk communication on man-made disasters and natural disasters with such characteristics as technical complexity, stigmatization and attribution of responsibility exposed about the man-made disasters. Based on the communication differences between these two types of disasters, a verification study has been designed using two types of data.
(8)Line 205:“There was variance between the volume of communication between events. What impact does the size and impact of an event have on risk communication and how does this impact your model given that man-made events usually have lower impact rates than natural events?”
Response:For any natural or man-made disaster, as long as the size of the event is very small, it will not likely to attract enough attention from the public. The model constructed in this paper has not taken this factor into account also because of this. Both disasters studied by this paper have generated huge social impact in China. While it would be more convincing to study man-made disasters and natural disasters of exactly the same scale, such cases would be rarely found in practice. Moreover, although man-made disasters have a lower impact rate, because of the roles of opinion leaders in man-made disasters, they will show more sensitivity to the size of the events.
(9)Line 230:“What is the relationship between transmission speed and information distortion, does volume of transmission from multiple sources also impact distortion?”
Response:This can present as another very crucial question. the inner logic should run like this: "the speed of information transmission has been slowed by the long information transmission chain, which points to greater possibility on rumor generation". The last sentence has been revised also accordingly as "longer information transmission path has reduced the speed of information transmission as well as increased possibility of generation of information distortion.”
(10)Line 248:“Really? How does this relate to opinion leaders?”
Response: This question has just hit the nail. The “opinion leaders” has been added to the sentence here, that is: “and the intention of the public to pass down information flown solely from trusted sources, or from those opinion leaders who hold discourse power in corresponding fields.”
(11)line 257:“Was there any analysis of the data set that demonstrated rates of information distortion or rumor that showed prevalence in long transmission vs short transmission to validate this sentence?”
Response: In view that the data collected in this paper is static data, that is, neither the dynamic changes of social media network structure of the event nor the correlation between information transmission chain and information distortion has been analyzed, a particular reference has been applied to prove this point (see Reference 39). Next, the research will focus on the dynamic changes of social media networks.
(12)line 261- line 263:“This is a contextual finding to where the event occurred, can this be validated elsewhere or is this model therefore limited only to this location?”
Response:First, the question on this finding in particular is greatly appreciated. Then, a supplementary explanation has since added to the analysis on the forwarding relationship of blogs of the "Sina Family" in this paper.
(13)485行: “No limitations of this research?”
Thank you for the reminder! The limitation has since been corrected as man-made disasters and natural disasters.
(14)Line 489-Line 497:“This belongs in the results section”
Response: The repeating part has since deleted with respect of similar content in the Result section.
(15)Line 512:“There can be conflicts of interest in reporting of the size and scale of impact, the needs of the affected population and the adequacy of response. This has been shown in many humanitarian crisis”.
Response: The original statement has been revised accordingly as " the factor of conflicts of interests between subjects plays weaker role for the natural disasters than man-made disasters”.
(16)Line 531:“This research did not investigate or show this”
The content which has not been investigated has since been deleted.
(17)Line 538:“Does it trust it more or does it show high use of access?”
Response: Compared with man-made disasters, more trust has been born by the public on information about natural disasters transmitted through social media. The corresponding parts have since been corrected as "only the accuracy of the information would be considered for risk communication on natural disasters. Because of the network’s high speed of information transmission, strong stability and high degree of public trust, the network can be used as the tool of information transmission and knowledge dissemination.”
Reference
Following reference have since been replaced and added.
Ano L.; Nancy M.; Paul L.H. Traditional and Social Media Coverage and Charitable Giving Following the 2010 Earthquake in Haiti. Prehosp Disaster Med 2012, 27(4), 319-324. Zhang, Y.; Su, Y.; Li, W. Rumor and authoritative information propagation model considering super spreading in complex social networks. Physic a-Statistical Mechanics and Its Applications. 2019, 506(15), 395-411.
